# Comparative Genomic Analysis of Key Oncogenic Pathways in Hepatocellular Carcinoma Among Diverse Populations

**DOI:** 10.3390/cancers17081309

**Published:** 2025-04-13

**Authors:** Cecilia Monge, Brigette Waldrup, Francisco G. Carranza, Enrique Velazquez-Villarreal

**Affiliations:** 1Center for Cancer Research, National Cancer Institute, Bethesda, MD 20892, USA; 2City of Hope, Beckman Research Institute, Department of Integrative Translational Sciences, Duarte, CA 91010, USA; 3City of Hope Comprehensive Cancer Center, Duarte, CA 91010, USA

**Keywords:** hepatocellular carcinoma, hepatic cancer, targeted therapy, cancer disparities, genetic mutations, precision medicine, TP53 pathway, WNT pathway, PI3K pathway, TGF-beta pathway, RTK/RAS pathway

## Abstract

Hepatocellular carcinoma (HCC) is a leading cause of cancer-related deaths, and Hispanic/Latino patients are often diagnosed at younger ages and more advanced stages than Non-Hispanic White patients. However, the genetic factors contributing to these differences are not well understood. This study analyzed genetic mutations in key cancer-related pathways to identify differences between these two populations. By examining publicly available data, researchers found that certain mutations in pathways involved in tumor growth and treatment response were more common in Hispanic/Latino patients. These findings provide new insights into the molecular differences in HCC between ethnic groups, which could help improve early detection and develop more effective, personalized treatments. Understanding these differences may also guide future research focused on reducing health disparities and improving outcomes for underrepresented populations.

## 1. Introduction

Liver cancer (LC) remains a significant global health burden, ranking as the sixth most common malignancy and the third leading cause of cancer-related deaths worldwide [1,2]. Since 2015, its incidence has steadily increased by 2% annually, contributing to a persistently low 5-year survival rate of only 22% [2]. The most prevalent form of LC is hepatocellular carcinoma (HCC), which accounts for approximately 90% of all primary liver cancer cases and primarily arises in the setting of chronic liver disease and cirrhosis [3]. HCC is a complex malignancy influenced by both intrinsic genetic factors and extrinsic environmental exposures, including viral hepatitis (HBV/HCV), metabolic dysfunction, obesity, diabetes, and dietary carcinogens [4]. In Hispanic/Latino (H/L) populations, a systematic review conducted in Texas identified diabetes, obesity, viral hepatitis, genetic predisposition, and environmental exposures as the leading risk factors for HCC development [5]. Alarmingly, H/L individuals face twice the mortality risk from HCC compared to their Non-Hispanic White (NHW) counterparts, highlighting stark disparities in disease burden and clinical outcomes [6,7]. Furthermore, H/L populations have the second-highest liver cancer incidence rate in the U.S. (15.1 cases per 100,000), surpassed only by American Indian/Alaska Native populations [8]. Despite these disparities, the molecular characterization of HCC in H/L populations remains vastly understudied, underscoring the need for genomic analyses that could uncover ethnicity-specific tumor biology and potential precision medicine interventions.

Molecular profiling studies of HCC have identified key oncogenic pathways [9]—RTK/RAS, TGF-beta, WNT, PI3K/AKT, and TP53—that play critical roles in tumor initiation, progression, and therapeutic resistance. The dysregulation of these pathways drives key hallmarks of cancer, including cell proliferation, survival, invasion, immune evasion, and chemoresistance. However, the genomic landscape of these pathways in H/L HCC patients remains largely unexplored, limiting the development of tailored therapeutic strategies for this population.

The RTK/RAS signaling pathway is a central regulator of cell growth, differentiation, and survival in HCC [10]. KRAS, NRAS, and BRAF mutations drive the constitutive activation of downstream MAPK and PI3K pathways, promoting tumor progression and drug resistance [11]. A study found that KRAS mutations were significantly more frequent in HCC patients with extrahepatic metastases, suggesting their role in disease aggressiveness [12]. Interestingly, HCC patients harboring RAS mutations have demonstrated positive responses to Refametinib in combination with Sorafenib, indicating the potential for RAS genotype-directed therapies [12,13,14,15,16,17,18,19,20]. However, the prevalence and clinical significance of RTK/RAS alterations in H/L HCC patients remain unclear.

The TGF-beta signaling pathway is a dual-function regulator in HCC, acting as a tumor suppressor in early-stage disease but promoting tumor progression, epithelial-to-mesenchymal transition (EMT), and immune evasion in advanced stages [21]. Genetic alterations in the TGF-beta pathway have been reported in 38% of HCC cases, with downstream effectors such as SMAD7 contributing to chemoresistance [9,22]. Given the aggressive nature of HCC in H/L populations, further exploration of TGF-beta pathway alterations may provide insights into disease progression and therapeutic response.

The WNT/β-catenin signaling pathway is a major driver of cell proliferation, tumor invasion, and immune evasion in HCC. CTNNB1 mutations, found in 30% of HCC cases, lead to aberrant WNT activation and have been linked to tumor growth, metastasis, and chemoresistance [23,24]. Additionally, AXIN1/2 mutations, present in 5–10% of cases, further contribute to WNT pathway dysregulation [25]. While activated WNT signaling has been associated with aggressive disease phenotypes in HCC, its role in H/L-specific tumor biology remains unknown [26].

The PI3K/AKT signaling pathway plays a pivotal role in cell survival, proliferation, and metabolic regulation in HCC. The activation of PI3K/AKT signaling, often through PIK3CA mutations and PTEN deletions, is associated with early recurrence, aggressive tumor behavior, and poor prognosis [27,28]. Notably, PIK3CB mutations have been linked to early-stage HCC recurrence, while PTEN loss has been observed in up to 53% of HCC tumors, resulting in uncontrolled PI3K/AKT activation [29,30,31]. However, the specific impact of PI3K/AKT pathway alterations in H/L HCC patients remains unknown, warranting further investigation.

The TP53 pathway, a critical regulator of genomic stability, apoptosis, and cell cycle control, is frequently altered in HCC. TP53 mutations occur in approximately 45% of HCC cases, leading to tumor progression, DNA repair defects, and therapy resistance [32,33]. Additionally, MDM2 overexpression and TP53 inactivation have been linked to chemoresistance, further complicating treatment outcomes [34,35]. Given the disparities in HCC prognosis among H/L populations, assessing TP53 pathway alterations could provide insight into potential ethnicity-specific therapeutic vulnerabilities.

Given the high burden of HCC in H/L populations and the limited molecular characterization of this disease [36] in this group, this study—building on prior health disparity research in other cancers [37,38,39]—aims to comprehensively analyze genomic alterations in the RTK/RAS, TGF-beta, WNT, PI3K, and TP53 pathways in HCC. We compare mutation frequencies between H/L and NHW patients and assess the prognostic impact of these alterations on overall survival. By integrating genomic and clinical data, this study seeks to uncover ethnicity-specific molecular differences that may contribute to HCC disparities. These findings could inform precision medicine approaches and help guide targeted therapeutic strategies to improve HCC outcomes in underrepresented populations.

## 2. Materials and Methods

We based our analysis on clinical and genomic data obtained from 15 HCC datasets accessed via the cBioPortal database. These datasets included studies classified under liver cancer, as well as data from the GENIE Cohort v17.0-public dataset. Following dataset selection, we applied a series of filtering criteria to refine our sample pool. Patients were included if they identified as H/L. This process resulted in three datasets meeting all criteria, comprising 69 H/L HCC patients. For NHW patients, 478 HCC patients were included using the same inclusion criteria but applied within this specific racial and ethnic group (Table 1 and Table 2). This study represents one of the largest comprehensive characterizations of TP53, WNT, PI3K, TGF-beta, and RTK/RAS pathway alterations in an underserved population, providing critical insights into the molecular disparities in HCC.

The study population was grouped into H/L and NHW categories according to self-identified ethnicity. We further stratified these groups based on the presence or absence of TP53, WNT, PI3K, TGF-beta, and RTK/RAS pathway alterations, enabling a detailed examination of the interactions between ethnicity and these molecular changes. Table 1 presents the number of patients included in the analysis of H/L and NHW HCC patients, with a total of 69 H/L patients and 478 NHW patients. This analysis evaluates the prevalence of TP53, WNT, PI3K, TGF-beta, and RTK/RAS pathway alterations by comparing H/L and NHW HCC patients. By integrating these stratifications, our study provides one of the most comprehensive characterizations of oncogenic pathway disruptions in an underserved population, offering valuable insights into potential molecular disparities and their implications for precision medicine in HCC.

To examine potential associations between ethnicity and pathway alterations, we employed Chi-square tests as part of our statistical analysis of categorical variables. This approach allowed us to assess whether certain molecular disruptions were more prevalent based on ethnicity, providing insights into patient heterogeneity and potential implications for treatment responses.

Given the imbalance in sample sizes between H/L and NHW patients, we employed statistical approaches appropriate for unequal group sizes. Mutation frequency comparisons between the two groups were conducted using chi-squared tests, which are robust for categorical data analysis even in the presence of unequal group sizes, provided that cell counts are adequate. All *p*-values are reported to ensure transparency, and findings with marginal significance were interpreted cautiously. While the current dataset represents one of the few publicly available resources with clinical and genomic data on H/L patients with HCC, we recognize the potential limitations in statistical power, particularly for detecting smaller effect sizes. As such, our analyses emphasize the identification of notable trends and candidate gene alterations that may warrant further validation in larger, more balanced cohorts in future studies.

As a note, for this analysis, we used publicly available HCC datasets. Only datasets that included both genomic mutation data and clearly annotated patient-level ethnicity information—specifically identifying individuals as H/L or NHW—were included. Datasets that lacked ethnicity data, had ambiguous classifications, or incomplete clinical annotations were excluded from this study to ensure accuracy in population stratification. This approach resulted in a final cohort of 547 patients, comprising 69 H/L and 478 NHW individuals. While the data were aggregated from multiple contributing institutions, we focused on high-level, consistently reported variables such as mutation status in key oncogenic pathways (RTK/RAS, TGF-beta, WNT, PI3K, TP53), gender, and overall survival.

The genes included in each oncogenic pathway analyzed in this study—RTK/RAS, TGF-beta, WNT, PI3K, and TP53—were obtained using the Integrated Pathway Mapper tool available through cBioPortal [40]. This tool relies on curated pathway definitions established by The Cancer Genome Atlas (TCGA) Research Network to standardize gene sets across major signaling pathways relevant to cancer. Specifically, the pathway definitions are based on the comprehensive analysis presented in a comprehensive systematic publication [41]. Utilizing this resource ensured that our pathway-level comparisons were grounded in biologically validated and widely accepted frameworks.

Kaplan–Meier survival analysis was employed to assess overall survival, focusing on the impact of TP53, WNT, PI3K, TGF-beta, and RTK/RAS pathway alterations. Survival curves were constructed to illustrate survival probabilities over time, with patients grouped by the presence or absence of pathway disruptions. The log-rank test was utilized to determine statistically significant differences between survival curves. Median survival times were calculated, accompanied by 95% confidence intervals to convey the precision of these estimates.

Kaplan–Meier survival analyses were conducted to evaluate overall survival differences associated with pathway-specific mutations between H/L and NHW patients. All survival analyses and associated statistical tests were performed using R statistical software (version 4.3.2). The survival package was used to fit Kaplan–Meier curves and perform log-rank tests, while the survminer package was used to generate visualizations. Patients were stratified by ethnicity and mutation status for each pathway, and survival curves were compared using the log-rank test to assess statistical significance.

To better characterize the biological relevance of pathway-specific alterations, we examined the nature of somatic mutations in genes that showed statistically significant or borderline significant differences between H/L and NHW patients. Specifically, we evaluated mutation types in TGFBR2, IGF1R, FGFR4, and INPP4B, focusing on alterations with potential functional impact. Mutation data that were extracted at the gene level from cBioPortal were just available from NHW populations and included nonsynonymous variants such as missense mutations, nonsense mutations, frame shift insertions and deletions, splice site mutations, and translation start site mutations. This classification was used to distinguish potentially deleterious mutations from those with uncertain or limited impact on protein function.

This comprehensive methodological approach provided an in-depth understanding of how specific pathway alterations may affect patient outcomes in HCC patients within the H/L population, offering new insights into the genomic landscape of an underrepresented group in liver cancer research.

## 3. Results

Using cBioPortal datasets that included ethnicity data, we assembled a cohort of 69 Hispanic/Latino (H/L) samples and 478 Non-Hispanic White (NHW) samples (Table 1). In terms of gender distribution, the H/L cohort was composed of 71.0% male (49 patients) and 29.0% female (20 patients), whereas the NHW cohort had a slightly lower proportion of male patients at 66.9% (320 patients) and a higher proportion of female patients at 33.1% (158 patients). Regarding tumor type, all patients in both cohorts were diagnosed with primary HCC (100%), with no cases of metastatic disease reported at the time of diagnosis. For ethnicity classification, the H/L cohort was further categorized into three subgroups: Spanish/Hispanic (62.3%), Spanish NOS/Hispanic NOS/Latino NOS (11.6%), and Hispanic or Latino (26.1%). In contrast, the NHW cohort was entirely composed of NHW patients (100%), ensuring a clear distinction in ethnic stratification. These demographic and clinical characteristics highlight the well-defined ethnic stratification and gender distribution between the cohorts, enabling a focused comparative genomic analysis of oncogenic pathway alterations in HCC among diverse populations.

The genomic comparison between H/L and NHW cohorts with HCC highlighted meaningful disparities in mutation rates, genomic instability, and the prevalence of driver mutations (Table 2). The median mutation count was slightly lower in the H/L cohort (six mutations, IQR: 4–46) compared to the NHW cohort (eight mutations, IQR: 4–50), though this difference did not reach statistical significance (*p* = 0.1087). Similarly, the median tumor mutational burden (TMB) was 3.78 mutations/Mb (IQR: 2.16–5.95) in H/L patients compared to 3.37 mutations/Mb (IQR: 1.73–4.92) in NHW patients, with no significant difference (*p* = 0.6563). These findings suggest that mutational burden does not significantly differ between H/L and NHW HCC patients, indicating potential similarities in underlying tumor biology across ethnic groups. In contrast, the median fraction of genome altered (FGA), a measure of chromosomal instability, was higher in H/L patients (0.22, IQR: 0.14–0.40) compared to NHW patients (0.19, IQR: 0.09–0.30), with a *p*-value of 0.05989, suggesting a potential trend toward increased structural alterations and copy number variations in the H/L cohort. This may indicate distinct genomic instability mechanisms in HCC among H/L individuals.

Furthermore, within the RTK/RAS pathway, a significant difference was observed in FGFR4 mutations, which were significantly more prevalent in the H/L cohort (4.3%) compared to the NHW cohort (0.6%), with a statistically significant *p*-value of 0.02906. FGFR4 alterations have been previously implicated in HCC progression, poor prognosis, and resistance to targeted therapies, suggesting that this ethnicity-associated difference may have important clinical and therapeutic implications. Overall, while mutational burden (TMB and mutation count) appeared similar between ethnic groups, H/L patients exhibited a trend toward higher chromosomal instability (FGA) and a significantly increased frequency of FGFR4 mutations, which may contribute to ethnic-specific differences in HCC progression, prognosis, and therapeutic response. Further investigation into the role of FGFR4 alterations in H/L HCC patients could provide critical insights for personalized treatment strategies and precision medicine approaches for this underrepresented population.

In our analysis of genetic alterations in HCC among H/L and NHW patients, we observed notable differences in key oncogenic pathways, though none reached statistical significance (Table 3). RTK/RAS pathway alterations were highly prevalent in both cohorts, with 30.4% of H/L patients and 31.0% of NHW patients exhibiting mutations (*p* = 1), suggesting a shared molecular profile in HCC development across both ethnic groups. This finding highlights the consistent role of RTK/RAS signaling in HCC progression, regardless of ethnicity. TGF-beta pathway alterations were less frequent in both groups, occurring in 2.9% of H/L patients compared to 5.4% of NHW patients (*p* = 0.56). While not statistically significant, this lower prevalence in H/L patients may indicate a potentially distinct role of TGF-beta signaling in tumor initiation or progression in this population. WNT pathway alterations were present in 46.4% of H/L patients and 45.2% of NHW patients (*p* = 0.9553), showing nearly identical distribution across ethnicities. Given the well-established role of WNT/β-catenin signaling in HCC tumorigenesis and metastasis, this suggests that HCC pathogenesis may be similarly influenced by WNT alterations in both groups. PI3K pathway alterations, which are implicated in tumor survival, immune evasion, and therapeutic resistance, were slightly more prevalent in H/L patients (21.7%) compared to NHW patients (17.4%) (*p* = 0.4728). Although not statistically significant, this trend suggests that H/L HCC tumors may exhibit more frequent PI3K signaling activation, potentially influencing treatment responses to PI3K/AKT/mTOR-targeted therapies. TP53 pathway alterations were observed in 40.6% of H/L patients and 34.9% of NHW patients (*p* = 0.4352). While not statistically significant, the higher prevalence of TP53 mutations in the H/L cohort is noteworthy, as TP53 inactivation is associated with increased genomic instability, tumor aggressiveness, and chemoresistance in HCC. Further studies are needed to assess whether ethnicity-specific TP53 alterations impact clinical outcomes or response to targeted therapies. Overall, while no statistically significant differences were identified between H/L and NHW patients for these pathway alterations, the observed trends suggest potential ethnicity-specific variations in oncogenic driver pathways. H/L patients showed a higher prevalence of PI3K and TP53 alterations, while RTK/RAS and WNT pathway mutations were nearly identical across both groups. These findings underscore the need for larger-scale studies to determine whether these molecular differences contribute to ethnic disparities in HCC incidence, progression, and therapeutic response.

Among H/L HCC patients, Kaplan–Meier analysis indicated that RTK/RAS pathway alterations were not associated with a statistically significant difference in overall survival (Figure 1). A total of 14 patients were in the altered group, while 69 patients were in the non-altered group, with both groups following nearly identical survival trajectories (*p* = 0.1). The overlapping confidence intervals highlight substantial variability in survival estimates, suggesting that RTK/RAS pathway alterations may not play a major prognostic role in this cohort. However, given the small sample size, additional studies with larger datasets are needed to validate these findings and explore potential subgroup-specific effects.

Similarly, TGF-beta pathway alterations did not show a statistically significant impact on survival outcomes in H/L HCC patients (Figure 1). The survival curve for the altered group showed an initial decline, followed by a plateau, with wide confidence intervals indicating high variability in survival estimates, likely due to the limited number of patients. These findings suggest that TGF-beta pathway alterations may not strongly influence prognosis in H/L HCC patients. However, given the biological significance of TGF-beta signaling in tumor progression and metastasis, further investigation with expanded cohorts is warranted to assess its potential clinical implications.

For WNT pathway alterations, Kaplan–Meier survival analysis showed no statistically significant differences (*p* = 0.53) in overall survival (Figure 1). A total of 20 patients were in the WNT pathway-altered group, with the remainder comprising the non-altered group. The survival curve for the altered group exhibited an initial decline, followed by stabilization, with wide confidence intervals suggesting considerable uncertainty in survival estimates. These findings indicate that WNT pathway alterations may not serve as a strong prognostic marker in H/L HCC patients. However, given the role of WNT signaling in tumor initiation and progression, further research with larger datasets is necessary to determine its clinical significance.

The PI3K pathway alterations followed a comparable trend to the other pathways analyzed, showing no statistically significant impact on survival outcomes (Figure 1). Among patients with PI3K pathway alterations, the survival curve displayed an early decline, followed by a plateau, with broad confidence intervals indicating high variability due to the small sample size (*p* = 0.82). These findings suggest that PI3K pathway alterations may not have a strong prognostic impact in H/L HCC patients. Further validation with larger datasets is needed to determine whether PI3K dysregulation influences tumor progression or therapy response in this population.

Kaplan–Meier analysis for TP53 pathway mutations (Figure 1) revealed that TP53 status did not significantly impact overall survival among H/L HCC patients (*p* = 0.55). Patients in the altered group exhibited an early decline in survival, followed by a plateau, while those in the non-altered group maintained a relatively stable trajectory. However, the wide and overlapping confidence intervals highlight substantial uncertainty in survival estimates, likely due to the small sample size. These findings suggest that TP53 pathway alterations may not serve as a strong prognostic marker in H/L HCC patients, emphasizing the need for larger studies to further investigate their clinical relevance.

Collectively, these results indicate that multiple pathway alterations in H/L HCC patients are not associated with significant differences in survival, suggesting limited prognostic value within this cohort. The broad confidence intervals and overlapping survival curves indicate substantial variability, likely influenced by limited sample sizes. These results underscore the need for larger, more diverse datasets to comprehensively evaluate the molecular drivers of HCC in H/L populations and determine their potential implications for disease progression, therapeutic response, and targeted treatment strategies.

Survival analysis using the Kaplan–Meier method examined RTK/RAS pathway alterations in NHW HCC patients and demonstrated no statistically significant difference in overall survival between those with and without RTK/RAS alterations (*p* = 0.55) (Appendix A). The survival trajectories of both groups remained closely aligned, with minimal separation in the curves, indicating that RTK/RAS pathway alterations may not serve as a strong prognostic marker in NHW HCC patients. However, the overlapping confidence intervals and variability in survival estimates highlight the need for further research with larger patient cohorts to determine whether RTK/RAS alterations interact with other molecular features to influence clinical outcomes.

We performed Kaplan–Meier survival analysis to evaluate TGF-beta pathway alterations in NHW HCC patients, and this analysis suggested a modest trend toward reduced survival in patients with alterations compared to those without (Appendix A). The TGF-beta-altered group (red curve) exhibited a gradual decline in survival probability, whereas the non-altered group (blue curve) showed a relatively more stable trajectory. However, this difference was not statistically significant (*p* = 0.24), and the wide confidence intervals surrounding the survival curves indicate a high degree of variability, likely due to the limited sample size in the altered group. Although these findings do not strongly support a major prognostic role for TGF-beta pathway alterations in NHW HCC, further investigations using larger datasets and refined subgroup analyses are needed to assess potential clinical relevance.

The use of Kaplan–Meier analysis allowed us to investigate WNT pathway alterations in NHW HCC patients, which similarly showed no statistically significant difference in survival outcomes (*p* = 0.2) (Appendix A). The WNT-altered group (red curve) exhibited a slight trend toward reduced survival compared to the non-altered group (blue curve), but the overlapping survival curves and wide confidence intervals suggest substantial variability in survival estimates. This variability is likely influenced by the small sample size in the altered group, reducing the statistical power to detect meaningful differences. While these findings do not provide strong evidence that WNT pathway alterations impact survival, further research with larger cohorts and molecular stratifications may help elucidate their potential prognostic role in NHW HCC.

Kaplan–Meier plots were utilized to explore survival differences related to PI3K pathway alterations in NHW HCC patients and showed a trend toward poorer overall survival, though it did not reach statistical significance (*p* = 0.09) (Appendix A). Patients with PI3K pathway alterations (red curve) exhibited a notable decline in survival probability compared to those without alterations (blue curve), suggesting that PI3K dysregulation may contribute to disease progression and worse prognosis in NHW HCC. The separation between the survival curves highlights the potential prognostic impact of PI3K pathway alterations. However, the wide confidence intervals in the altered group indicate variability in survival estimates, likely due to the smaller sample size. These findings underscore the need for further studies to explore the biological mechanisms underlying PI3K pathway alterations and their potential role as therapeutic targets in NHW HCC.

Lastly, the Kaplan–Meier survival analysis for TP53 pathway alterations in NHW HCC patients revealed no significant difference in survival between those with and without TP53 alterations (*p* = 0.43) (Appendix A). The survival trajectories of both groups were largely similar, and the overlapping confidence intervals suggest considerable variability in survival estimates. These findings indicate that TP53 pathway alterations may not serve as a major prognostic factor in NHW HCC. However, given the well-established role of TP53 in tumor progression, genomic instability, and therapy resistance, additional studies with larger sample sizes and more refined molecular analyses are warranted to assess whether TP53 mutations contribute to survival disparities in specific HCC subtypes or therapeutic responses.

These observations suggest that survival outcomes among NHW HCC patients may be influenced by alterations in specific signaling pathways. While RTK/RAS, TGF-beta, WNT, and TP53 pathway alterations did not significantly impact survival, PI3K pathway alterations showed a trend toward poorer prognosis, though the association did not reach statistical significance. The overlapping survival curves and wide confidence intervals for most pathways suggest that sample size limitations and molecular heterogeneity may obscure potential prognostic differences. These results underscore the importance of comprehensive genomic analyses and larger cohort studies to better define the clinical significance of these molecular alterations. Further investigation into PI3K pathway dysregulation is particularly warranted, as it may play a role in disease progression and serve as a potential therapeutic target in NHW HCC patients.

Alteration patterns in RTK/RAS pathway genes were assessed across H/L and NHW HCC patients to explore ethnicity-based differences (Appendix A). Most genes in the RTK/RAS pathway exhibited low mutation frequencies across both groups, with no statistically significant differences for most genes. However, FGFR4 mutations were significantly more prevalent in H/L patients (4.3%) compared to NHW patients (0.6%) (*p* = 0.02906), suggesting a possible ethnicity-specific molecular alteration. Other receptor tyrosine kinase (RTK) genes, including EGFR, ERBB2, ERBB4, MET, and IGF1R, were more frequently altered in H/L patients, though these differences did not reach statistical significance (*p* > 0.05). Notably, ERBB4 mutations were observed in 5.8% of H/L patients compared to 2.3% of NHW patients (*p* = 0.1077), and IGF1R mutations were detected in 7.2% of H/L patients versus 2.9% of NHW patients (*p* = 0.07798), indicating a possible trend toward increased RTK pathway activation in H/L patients. Within the RAS signaling branch, NRAS mutations were detected at a higher frequency in H/L patients (2.9%) compared to NHW patients (1.3%), though this difference was not significant (*p* = 0.2669). In contrast, KRAS, HRAS, and BRAF mutations were rare in both populations. Other RTK/RAS-related genes, including PDGFRA, RET, ROS1, and PTPN11, showed low mutation rates without significant differences. These findings suggest that FGFR4 mutations may be an ethnicity-associated molecular feature in H/L HCC patients, warranting further research into their functional implications and potential as a therapeutic target. Additionally, the higher mutation rates in genes like ERBB4 and IGF1R (albeit not statistically significant) suggest ethnic variability in RTK/RAS pathway activation, highlighting the need for larger studies to validate these observations.

The TGF-beta pathway analysis revealed that TGFBR2 mutations were more frequent in H/L HCC patients (2.9%) compared to NHW patients (0.4%), showing a borderline statistical significance (*p* = 0.07951, Appendix A). This suggests a potential ethnicity-related disparity in TGFBR2 alterations, which could influence HCC progression and response to therapy. Other TGF-beta pathway-related genes, including ACVR2A, ACVR2B, SMAD2, SMAD3, SMAD4, and TGFBR1, exhibited low mutation rates in both groups, with no statistically significant differences (*p* > 0.05). Notably, SMAD4 mutations were absent in H/L patients but present in 0.8% of NHW patients, which is consistent with prior studies suggesting its role as a tumor suppressor frequently altered in HCC. These findings indicate that TGFBR2 mutations may be more prevalent in H/L patients, emphasizing the need for further investigation into their functional impact on TGF-beta signaling and potential targeted therapy options.

In the WNT pathway, AXIN1 mutations were more frequent in H/L HCC patients (8.7%) compared to NHW patients (3.8%), though the difference did not reach statistical significance (*p* = 0.12, Appendix A). Similarly, APC mutations were slightly higher in H/L patients (5.8%) than in NHW patients (4.8%), but this difference was also not statistically significant (*p* = 0.7642). Other WNT pathway-related genes, including CTNNB1, AXIN2, and TCF7L2, exhibited similar alteration rates between the two groups, with no significant differences detected. CTNNB1 mutations were highly prevalent in both groups (31.9% in H/L vs. 32.4% in NHW, *p* = 1), highlighting its role as a shared oncogenic driver in HCC. Interestingly, DKK2 and TCF7L1 mutations were exclusive to H/L patients, whereas RNF43, LRP5, and LRP6 mutations were only observed in NHW patients, albeit at very low frequencies. These findings suggest that AXIN1 and APC mutations may be slightly more common in H/L patients, while other WNT pathway alterations appear comparable across ethnicities. Additional research is warranted to elucidate the functional impact of these mutations on WNT signaling, HCC progression, and treatment response.

The PI3K pathway analysis revealed that PTEN mutations were more frequent in H/L HCC patients (4.3%) compared to NHW patients (1.5%), though this difference was not statistically significant (*p* = 0.1202, Appendix A). Similarly, INPP4B mutations were detected in 4.3% of H/L patients and 1.0% of NHW patients, showing a trend toward increased prevalence in H/L patients (*p* = 0.06749). However, PIK3CA mutations, a key oncogenic driver, were observed at similar frequencies in both groups (2.9% in H/L vs. 1.9% in NHW, *p* = 0.6374), suggesting no clear ethnicity-specific pattern in this gene’s mutation rate. Other PI3K pathway-related genes, including PIK3R1, PIK3R2, AKT2, AKT3, and RICTOR, had low or absent mutation rates in both populations. Notably, MTOR mutations were present at identical rates in H/L and NHW patients (2.9% in both groups, *p* = 1), suggesting a shared oncogenic mechanism in HCC development. These findings indicate a potential trend toward increased PTEN and INPP4B mutations in H/L patients, but, given low mutation frequencies and a lack of statistical significance, larger studies are needed to assess the role of PI3K pathway alterations in HCC across ethnic groups.

Lastly, the TP53 pathway analysis showed that TP53 mutations were more common in H/L HCC patients (34.8%) than in NHW patients (27.4%), though this difference was not statistically significant (*p* = 0.2592, Appendix A). Similarly, MDM2 mutations were observed in 1.4% of H/L patients but were absent in NHW patients, though the low frequency limited statistical significance (*p* = 0.1261). Conversely, ATM mutations were more frequent in NHW patients (5.0%) compared to H/L patients (1.4%), but this difference was not statistically significant (*p* = 0.3482). Other TP53 pathway genes, including CDKN2A, CHEK2, and RPS6KA3, displayed similar alteration rates across both groups, with no statistically significant differences. These findings suggest that, while TP53 mutations may be slightly more prevalent in H/L patients, other genes in this pathway do not show substantial ethnic variability. Further studies with larger cohorts are necessary to evaluate whether these differences impact tumor biology, disease progression, or therapeutic response in HCC.

To further explore the potential functional impact of the observed gene alterations, we analyzed the nature of mutations in genes that showed significant or borderline significant differences between Hispanic/Latino and Non-Hispanic White (NHW) patients. As shown in Appendix A, mutations in TGFBR2, IGF1R, FGFR4, and INPP4B among NHW patients were primarily classified as nonsynonymous, with notable variation in mutation types across genes. TGFBR2 mutations in NHW patients were evenly distributed between missense mutations (50%) and splice region variants (50%), suggesting the potential disruption of gene splicing and protein function. IGF1R mutations consisted predominantly of missense mutations (80%), with smaller proportions of frame shift deletions (7%), nonsense mutations (7%), and splice site alterations (7%). Both FGFR4 and INPP4B mutations were exclusively missense in nature (100%), indicating amino acid substitutions that may impact protein activity without introducing premature stop codons or frame shifts. This mutation-type analysis supports the interpretation that these genes may contribute to pathway dysregulation in HCC through distinct mechanisms of functional alteration.

The data point to potential ethnic differences in the molecular landscape of HCC, especially with increased mutation frequencies in FGFR4, TGFBR2, AXIN1, PTEN, and TP53 among H/L individuals. However, most comparisons did not reach statistical significance, underscoring the need for larger, well-powered studies to confirm these observations. The higher prevalence of FGFR4 mutations in H/L patients suggests potential differences in RTK/RAS pathway activation, while increased TGFBR2 alterations may indicate ethnicity-specific variations in TGF-beta signaling. Similarly, AXIN1 and PTEN mutations showed a trend toward higher frequencies in H/L patients, suggesting possible differences in WNT and PI3K pathway dysregulation. Although TP53 mutations were more prevalent in H/L patients, the overall mutation rates across the TP53 pathway were comparable between groups. These findings emphasize the importance of ethnicity-stratified genomic analyses to better understand HCC molecular disparities and inform precision medicine approaches and targeted therapies for underrepresented populations.

## 4. Discussion

HCC is a major global health concern, with a growing incidence and persistently poor prognosis, particularly among H/L populations. Despite the high disease burden in this group, genomic studies have largely focused on Asian and NHW populations, leaving critical gaps in understanding the molecular underpinnings of HCC in H/L patients. This study aimed to comprehensively analyze the genomic alterations in key oncogenic pathways, including RTK/RAS, TGF-beta, WNT, PI3K, and TP53, to assess potential ethnicity-specific differences in mutation prevalence and their prognostic implications.

Ethnicity-Specific Pathway Alterations

Our findings suggest that, while the overall mutation burden was similar between H/L and NHW patients, certain oncogenic pathway alterations exhibited ethnicity-associated trends. Within the RTK/RAS pathway, FGFR4 mutations were significantly more frequent in H/L patients (4.3%) compared to NHW patients (0.6%) (*p* = 0.02906), suggesting a potential ethnicity-associated driver alteration in HCC. Given that FGFR4 activation has been linked to aggressive tumor phenotypes and therapeutic resistance, its higher prevalence in H/L patients raises important questions about its role in HCC progression and targeted therapy response. Although other RTK/RAS genes, including EGFR, ERBB2, ERBB4, MET, and IGF1R, exhibited higher alteration rates in H/L patients, these differences did not reach statistical significance, emphasizing the need for further validation in larger cohorts.

In the TGF-beta pathway, TGFBR2 mutations were more prevalent in H/L patients (2.9%) than NHW patients (0.4%), showing borderline statistical significance (*p* = 0.07951). As a key receptor in TGF-beta signaling, TGFBR2 loss-of-function mutations can shift TGF-beta from a tumor-suppressive to a pro-oncogenic role, promoting immune evasion and metastasis. This finding suggests that H/L patients may have distinct alterations in TGF-beta signaling, warranting further research into its functional implications for tumor progression and potential therapeutic targeting.

Within the WNT pathway, AXIN1 mutations were more common in H/L patients (8.7%) than in NHW patients (3.8%), though this difference did not reach statistical significance (*p* = 0.12). Since AXIN1 is a negative regulator of WNT/β-catenin signaling, its mutation can lead to aberrant pathway activation, contributing to tumor growth and metastasis. Similarly, APC mutations were slightly higher in H/L patients (5.8% vs. 4.8% in NHW), suggesting a possible trend toward increased WNT dysregulation in this population. However, CTNNB1 mutations, a major driver of WNT activation, were highly prevalent in both groups (31.9% H/L vs. 32.4% NHW, *p* = 1), indicating that β-catenin pathway activation is a shared oncogenic feature of HCC across ethnicities.

For the PI3K pathway, PTEN mutations were more frequent in H/L patients (4.3%) than NHW patients (1.5%), though the difference was not statistically significant (*p* = 0.1202). PTEN is a key tumor suppressor that negatively regulates PI3K/AKT signaling, and its inactivation is associated with tumor progression, therapy resistance, and immune evasion. Similarly, INPP4B mutations were observed in 4.3% of H/L patients vs. 1.0% of NHW patients (*p* = 0.06749), suggesting potential ethnicity-specific differences in PI3K pathway dysregulation. However, PIK3CA mutations were found at similar rates in both groups (2.9% in H/L vs. 1.9% in NHW, *p* = 0.6374), indicating that PI3K activation mechanisms may vary across ethnicities. Further functional studies are needed to determine whether the observed trends in PTEN and INPP4B mutations translate into differential tumor behavior or therapeutic responses.

An important consideration in this study is the reliance on self-reported ethnicity, as captured in the publicly available cBioPortal and GENIE datasets, which do not include the individual-level genotype data required for performing ancestry inference. While self-identified ethnicity provides valuable sociocultural and epidemiological context, it may not fully reflect an individual’s genetic ancestry. Consequently, the lack of ancestry-based classification may limit the precision of our population stratification and the interpretation of ethnicity-specific genomic differences. In other cancer studies involving underrepresented populations, researchers have increasingly adopted both global genomic ancestry approaches to improve the resolution of disparity research and better capture the genetic diversity within populations [37,38,42]. Building on this progress, future analyses will benefit from incorporating ancestry inference when datasets with sufficient genomic granularity become available.

Lastly, the TP53 pathway analysis revealed that TP53 mutations were more frequent in H/L patients (34.8%) than in NHW patients (27.4%), though this difference was not statistically significant (*p* = 0.2592). Given the role of TP53 in genomic stability, apoptosis, and therapy resistance, its slightly higher prevalence in H/L patients suggests a possible contributor to tumor progression and treatment outcomes in this group. MDM2 mutations were detected in 1.4% of H/L patients but were absent in NHW patients, while ATM mutations were more common in NHW patients (5.0% vs. 1.4% in H/L, *p* = 0.3482). These results indicate that, while TP53 inactivation is a hallmark of HCC across ethnicities, secondary alterations in DNA damage response pathways may vary between populations.

Prognostic Implications of Pathway Alterations

Kaplan–Meier survival analyses revealed no statistically significant differences in overall survival for RTK/RAS, TGF-beta, WNT, PI3K, or TP53 pathway alterations in H/L patients. The overlapping confidence intervals and broad variability in survival estimates suggest that pathway alterations alone may not be strong prognostic indicators in this population. This could be due to heterogeneity in tumor biology, immune response, or environmental factors that modulate survival outcomes.

In contrast, PI3K pathway alterations in NHW patients were associated with a trend toward poorer survival (*p* = 0.09), suggesting that PI3K dysregulation may play a greater prognostic role in NHW HCC patients. Previous studies have shown that PI3K/AKT/mTOR pathway activation is linked to tumor progression, therapy resistance, and poor prognosis, particularly in tumors with high metabolic demand. The observed survival differences between H/L and NHW patients suggest that ethnicity-specific tumor microenvironment interactions may influence PI3K pathway activation and therapeutic vulnerability.

Similarly, TGF-beta pathway alterations in NHW patients were associated with a modest decline in survival (*p* = 0.24), whereas no significant differences were observed in H/L patients. Given that TGF-beta signaling can promote immune evasion and metastasis in advanced HCC, these findings raise important questions about whether ethnicity-specific immune responses may modulate the impact of TGF-beta pathway dysregulation on survival outcomes.

Clinical and Translational Implications

The observed trends in FGFR4, TGFBR2, AXIN1, PTEN, and TP53 mutations in H/L patients suggest that certain oncogenic pathways may be preferentially dysregulated in this population, although statistical significance was not reached in most cases. These findings underscore the need for larger, ethnicity-stratified genomic studies to validate these observations and explore their potential therapeutic implications. Given the higher prevalence of FGFR4 mutations in H/L patients, FGFR inhibitors may warrant further investigation as targeted therapy options for this group. Similarly, the trend toward increased TGFBR2 mutations suggests that H/L patients may benefit from therapies targeting TGF-beta signaling, particularly in the context of an immune checkpoint blockade.

The lack of significant survival differences in H/L patients across oncogenic pathways suggests that tumor biology, host immune response, and treatment access disparities may contribute to HCC outcomes in this group. Future studies should integrate multi-omics analyses, including transcriptomics [38], proteomics [39] and using single-cell sequencing technologies [43,44], to uncover potential regulatory mechanisms that may drive HCC progression in H/L patients. Additionally, clinical trials should ensure the adequate representation of H/L patients to assess whether ethnicity-specific genomic alterations influence treatment response and survival.

An acknowledged limitation of this study is the imbalance in sample sizes between the H/L and NHW patient groups, with 69 and 478 individuals, respectively. This discrepancy reflects the broader underrepresentation of H/L populations in publicly available genomic datasets for HCC, which poses challenges for conducting statistically robust disparity-focused research. Despite the limited sample size, we identified several potentially meaningful differences in pathway-specific alterations, particularly in genes such as FGFR4, IGF1R, INPP4B, and TGFBR2. While larger effect sizes can still yield statistically significant findings in smaller cohorts, we recognize that smaller effect sizes may be underpowered in this context. We have therefore interpreted borderline associations with caution and reported all *p*-values transparently. Future studies with larger, more balanced cohorts and expanded genomic data for underrepresented populations will be essential to validate and extend these findings. Our work highlights the importance of continued efforts to increase diversity in cancer genomic research and the urgent need for more inclusive datasets to enable equitable precision medicine.

We acknowledge that the data used in this study were aggregated from multiple sources through cBioPortal and GENIE, which may introduce heterogeneity in clinical evaluations, data annotation, and treatment protocols across contributing institutions. While this reflects a limitation inherent in working with large, multi-institutional datasets, we addressed it by including only those studies that provided both genomic and clearly defined ethnicity data. Additionally, we limited our analysis to consistently reported clinical and molecular features to reduce variability across datasets. We have also clearly outlined our inclusion criteria and data sources in the Methods Section. Despite these limitations, our study represents one of the first attempts to explore ethnicity-specific oncogenic pathway alterations in HCC using available public data, and it highlights the urgent need for more harmonized and inclusive genomic resources to support cancer disparity research.

It is important to note that several genes within the oncogenic pathways analyzed in this study—particularly those in the TGF-beta and PI3K pathways—are known to exhibit context-dependent functions, acting as either tumor suppressors or oncogenes depending on factors such as tumor stage, cellular context, and the broader mutational landscape. While we used established pathway definitions from the TCGA-based Integrated Pathway Mapper in cBioPortal to group genes for analysis, this pathway-level approach does not fully capture the dynamic and sometimes dual roles of individual genes in tumor progression. As such, the biological and clinical implications of these mutations should be interpreted with caution. Future studies incorporating functional validation and more granular stratification by tumor stage and molecular context will be essential to fully understand the role of these genes in hepatocellular carcinoma, particularly in the context of ethnicity-specific differences.

The analysis of mutation types in genes with significant or borderline significance provided additional insight into their potential biological impact on HCC pathogenesis. As shown in Appendix A, mutations in TGFBR2, IGF1R, FGFR4, and INPP4B among NHW patients were predominantly nonsynonymous, with a majority being missense mutations. The presence of splice region alterations in TGFBR2 and the mixture of mutation types in IGF1R (including missense, nonsense, and frame shift deletions) suggest the functional disruption of key signaling components. In contrast, mutations in FGFR4 and INPP4B were exclusively missense, potentially affecting protein function through amino acid substitutions without introducing frameshifts or truncations. These patterns highlight the variability in the nature of mutations across pathway components and reinforce the importance of considering mutation type—not just frequency—when evaluating their contribution to oncogenic signaling. Future studies incorporating functional assays and in-depth structural modeling will be valuable for validating the impact of these specific alterations and understanding their roles in driving ethnic disparities in HCC.

An important consideration in interpreting our findings is the potential influence of a population-specific genetic background on the observed differences in mutation frequencies. While our analysis focused exclusively on somatic mutations identified in HCC tumors, we acknowledge that some of the mutations more frequently observed in H/L patients may reflect underlying germline variation or broader population-specific genetic architecture rather than being direct oncogenic drivers. Unfortunately, the publicly available datasets used in this study do not include matched germline data or allele frequency information from non-cancer H/L populations, which limits our ability to fully distinguish between cancer-specific mutations and background variants. Future studies that incorporate both somatic and germline data across diverse populations will be essential to validate the cancer-specific relevance of these alterations.

Another important consideration in this study is the absence of data on potential confounding variables such as alcohol consumption, access to healthcare, lifestyle behaviors, and socioeconomic status. These factors are well-established contributors to HCC risk, progression, and outcomes, and they may partially account for the disparities observed between ethnic groups. However, the publicly available datasets utilized in this analysis do not contain comprehensive clinical or social determinant information, which restricted our ability to control for these influences. Consequently, while our results reveal differences in somatic mutation patterns between H/L and NHW patients, they should be interpreted in the context of broader, unmeasured environmental and societal variables. Future research efforts that integrate genomic data with robust clinical and sociodemographic information will be critical for disentangling the complex factors driving HCC disparities and for advancing equitable precision medicine.

## 5. Conclusions

This study provides one of the first comparative genomic analyses of oncogenic pathway alterations in H/L and NHW HCC patients. While most pathway alterations were similar between groups, FGFR4, TGFBR2, AXIN1, PTEN, and TP53 mutations showed ethnicity-associated trends that may warrant further investigation. The lack of statistically significant survival differences in H/L patients suggests that additional molecular and environmental factors may contribute to HCC disparities in this population. These findings emphasize the need for larger, prospective studies focusing on H/L populations to fully characterize the molecular landscape of HCC and inform ethnicity-specific precision medicine approaches.

## Figures and Tables

**Figure 1 cancers-17-01309-f001:**
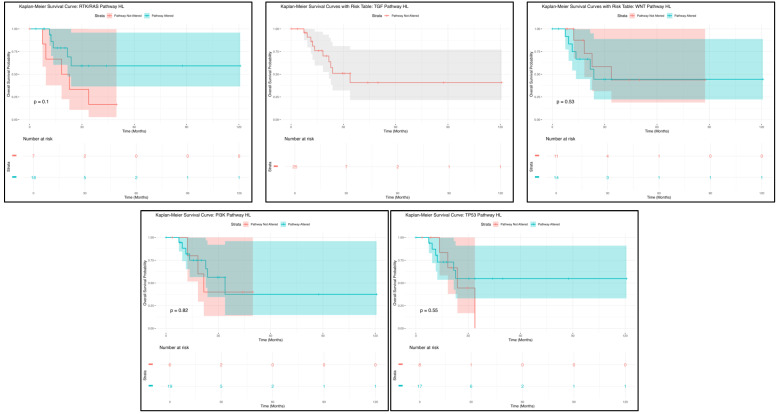
Kaplan–Meier overall survival curves for Hispanic/Latino (H/L) hepatocellular carcinoma (HCC) patients, stratified by the presence or absence of RTK/RAS (**upper left**), TGF-beta (**upper middle**), WNT (**upper right**), PI3K (**lower left**), and TP53 (**lower right**) pathway alterations.

**Table 1 cancers-17-01309-t001:** Patient demographics and clinical characteristics of the Hispanic/Latino (H/L) and Non-Hispanic White (NHW) hepatocellular carcinoma (HCC) cohorts.

Clinical Feature	H/L Cohort	NHW Cohort
	*n* (%)	*n* (%)
Gender		
Male	49 (71.0%)	320 (66.9%)
Female	20 (29.0%)	158 (33.1%)
Tumor Type		
Primary	69 (100.0%)	478 (100.0%)
Metastatic	0 (0.0%)	0 (0.0%)
Ethnicity		
Spanish/Hispanic	43 (62.3%)	0 (0.0%)
Spanish NOS; Hispanic NOS; Latino NOS	8 (11.6%)	0 (0.0%)
Hispanic or Latino	18 (26.1%)	0 (0.0%)
Non-Spanish; Non-Hispanic *	0 (0.0%)	478 (100.0%)

* Ethnicity classification was based on self-reported data available from public databases. All individuals categorized as NHW were recorded as White in the race field and “Non-Spanish; Non-Hispanic” in the ethnicity field. No individuals in the NHW group reported any Hispanic, Latino, or Spanish ethnicity.

**Table 2 cancers-17-01309-t002:** Ethnicity-associated differences in clinical features between Hispanic/Latino (H/L) and Non-Hispanic White (NHW) hepatocellular carcinoma (HCC) cohorts.

Clinical Feature	H/L Samples *n* (%)	NHW Samples *n* (%)	*p*-Value
Median Mutation Count (IQR) *	6 (4–46)	8 (4–50)	0.1087
Median TMB (IQR) **	3.78 (2.16–5.95)	3.37 (1.73–4.92)	0.6563
Median FGA (IQR) ***	0.22 (0.14–0.40)	0.19 (0.09–0.30)	0.05989
Oncotree Code			
HCC	69 (100.0%)	478 (100.0%)	NA
FGFR4 Mutation			
Present	3 (4.3%)	3 (0.6%)	0.02906
Absent	66 (95.7%)	475 (99.4%)	
* HL NA: 4, NHW NA: 22			
** HL NA: 61, NHW NA: 324			
*** HL NA: 21, NHW NA: 105			

* NA values for Median Mutation Count: H/L = 4, NHW = 22. ** NA values for Median TMB: H/L = 61, NHW = 324. *** NA values for Median FGA: H/L = 21, NHW = 105. These indicate the number of samples with missing data for each respective variable.

**Table 3 cancers-17-01309-t003:** Rates of TGF-beta, RTK/RAS, WNT, PI3K, and TP53 pathway alterations among Hispanic/Latino (H/L) and Non-Hispanic White (NHW) hepatocellular carcinoma (HCC) patients.

	H/L Samples *n* (%)	NHW Samples *n* (%)	*p*-Value
RTK/RAS Alterations Present	21 (30.4%)	148 (31.0%)	1
RTK/RAS Alterations Absent	48 (69.6%)	330 (69.0%)	
TGF-beta Alterations Present	2 (2.9%)	26 (5.4%)	0.56
TGF-beta Alterations Absent	67 (97.1%)	452 (94.6%)	
WNT Alterations Present	32 (46.4%)	216 (45.2%)	0.9553
WNT Alterations Absent	37 (53.6%)	262 (54.8%)	
PI3K Alterations Present	15 (21.7%)	83 (17.4%)	0.4728
PI3K Alterations Absent	54 (78.3%)	395 (82.6%)	
TP53 Alterations Present	28 (40.6%)	167 (34.9%)	0.4352
TP53 Alterations Absent	41 (59.4%)	311 (65.1%)	

## Data Availability

All data used in the present study are publicly available at https://www.cbioportal.org/ (accessed on 14 February 2025) and https://genie.cbioportal.org (accessed on 14 February 2025). Additional data can be provided upon reasonable request to the authors.

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
