# Peer review of "Comparative Genomic Analysis of Key Oncogenic Pathways in Hepatocellular Carcinoma Among Diverse Populations"

_cancers, 2025, doi:10.3390/cancers17081309_

Round 1
Reviewer 1 Report
Comments and Suggestions for Authors
The present study titled "Comparative genomic analysis of key oncogenic pathways in hepatocellular carcinoma among diverse populations" by Monge et al is based on statistical analysis of publicly available data to reveal the ethnicity specific mutational differences. I have following comments:
- As ethnicity lies in the core of this analysis therefore it becomes crucial to determine the ethnicity by ancestry analysis rather than relying on self declared ethnicity. Therefore I request the authors to perform and ancestry analysis to include patient population based on ancestry analysis. This will provide more accurate understanding of someone's genetic heritage.
- In this analysis I noticed that one group H/L has only 69 patients while other group NHW has 478 patients. How did the authors handle these imbalanced datasets. The analysis performed here can be valid if the effect size of the studied parameters is large. However, for smaller effect size authors will need careful selection of statistical parameters that need to be explained in methods section.
- Some minor comments are
- Table 1 "Non Spanish non hispanic need to be clarified. Which ethnicity they belong to ?
- The overall manuscript need language corrections to make sure the follow of the content is clear.
- Regarding the ithenticate report, it shows 38% similarity which is way above the threshold. Kindly revise the manuscript to remove the similarity.
English needs improvement.
Author Response
Reviewer 1 Comments - Attached Word file: Response_Reviewer_1_Comments_040425
We are pleased to re-submit our manuscript titled “Comparative Genomic Analysis of Key Oncogenic Pathways in Hepatocellular Carcinoma Among Diverse Populations,” and we believe it will be of strong interest to the scientific community. We have carefully addressed all reviewer comments and further emphasized the significance of this cancer disparity. Our study presents one of the first ethnicity-focused genomic analyses of HCC, leveraging one of the few publicly available datasets with sufficient representation of Hispanic/Latino (H/L) and Non-Hispanic White (NHW) patients. By examining mutation frequencies in critical oncogenic pathways—RTK/RAS, TGF-Beta, WNT, PI3K, and TP53—we identify key molecular differences that may contribute to the observed disparities in HCC incidence and progression. Notably, our findings highlight the enrichment of FGFR4, IGF1R, INPP4B, and TGFBR2 alterations among H/L patients, providing novel insights that could inform precision medicine strategies and enhance clinical outcomes in underrepresented populations. Thank you for your thoughtful review and consideration. Please find our detailed responses below, with corresponding revisions marked in blue font and highlighted in yellow in the re-submitted Word document.
Thank you very much for taking the time to review this manuscript. Please find the detailed responses below and the corresponding revisions wrote in blue font and highlighted in yellow in the re-submitted Word file.
Reviewer 1 acknowledged the relevance of our study titled “Comparative Genomic Analysis of Key Oncogenic Pathways in Hepatocellular Carcinoma Among Diverse Populations,” recognizing the value of our statistical analysis of publicly available data to uncover ethnicity-specific mutational differences. This feedback highlights the contribution of our work to the growing body of research focused on molecular disparities in cancer and supports the importance of leveraging existing genomic resources to identify underexplored differences across diverse populations. The reviewer’s recognition affirms the potential impact of our findings and further supports the manuscript’s suitability for publication in Cancers.
Reviewer 1 writes:
“The present study titled "Comparative genomic analysis of key oncogenic pathways in hepatocellular carcinoma among diverse populations" by Monge et al is based on statistical analysis of publicly available data to reveal the ethnicity specific mutational differences.”
We thank Reviewer 1 for their thoughtful and constructive feedback. We are pleased that the reviewer recognized the relevance of our study and its focus on uncovering ethnicity-specific mutational differences in hepatocellular carcinoma through statistical analysis of publicly available data. We appreciate the acknowledgment of the importance of this approach in addressing gaps in our understanding of molecular disparities across diverse populations. We have carefully reviewed the comment and ensured that the manuscript clearly communicates the significance of our findings and the rigor of our analytical methods to further strengthen its contribution to the field.
Comment 1:
“As ethnicity lies in the core of this analysis therefore it becomes crucial to determine the ethnicity by ancestry analysis rather than relying on self declared ethnicity. Therefore I request the authors to perform and ancestry analysis to include patient population based on ancestry analysis. This will provide more accurate understanding of someone's genetic heritage.”
Response:
We thank Reviewer 1 for this insightful comment. We fully agree that determining ethnicity through genomic ancestry analysis offers a more accurate understanding of an individual’s genetic background and is particularly important in studies focused on cancer disparities. However, due to the limitations of the publicly available datasets used in our study (cBioPortal and GENIE), individual-level genotype data required for global or local ancestry inference are not accessible. As a result, we were constrained to rely on self-reported ethnicity as recorded in these databases.
We would like to note that our group is actively engaged in advancing genomic ancestry research and has pioneered ancestry inference methods, with several studies in this area [i.e., PMIDs: 39999309, 39606335, 40034762] that were included in the references 37, 38, and 42. To address the reviewer’s concern, we have added a paragraph to the Discussion section explicitly acknowledging this limitation and citing our prior work on genomic ancestry analysis in this underrepresented population to underscore our awareness of its importance and our commitment to integrating such approaches when data permits.
The revised paragraph added to the Discussion section, lines 585-596, now reads as follows:
“An important consideration in this study is the reliance on self-reported ethnicity, as captured in the publicly available cBioPortal and GENIE datasets, which do not include the individual-level genotype data required for performing ancestry inference. While self-identified ethnicity provides valuable sociocultural and epidemiological context, it may not fully reflect an individual’s genetic ancestry. Consequently, the lack of ancestry-based classification may limit the precision of our population stratification and the interpretation of ethnicity-specific genomic differences. In other cancer studies involving underrepresented populations, researchers have increasingly adopted both global and local genomic ancestry approaches to improve the resolution of disparities research and better capture the genetic diversity within populations [37, 38, 42]. Building on this progress, future analyses will benefit from incorporating ancestry inference when datasets with sufficient genomic granularity become available.”
We thank the reviewer for these constructive comments. We believe that this clarification, supported by relevant references, enhances the contextual understanding and provides important insights into the complexities of population stratification.
Comment 2:
“In this analysis I noticed that one group H/L has only 69 patients while other group NHW has 478 patients. How did the authors handle these imbalanced datasets. The analysis performed here can be valid if the effect size of the studied parameters is large. However, for smaller effect size authors will need careful selection of statistical parameters that need to be explained in methods section.”
Response:
We thank the reviewer for this important and insightful comment. We acknowledge the imbalance in sample sizes between H/L patients (n = 69) and NHW patients (n = 478) in our analysis. This disparity reflects the broader underrepresentation of H/L individuals in publicly available genomic and clinical datasets, particularly for HCC. Our study utilizes one of the very few resources currently available that include both genomic and clinical data for H/L patients with HCC, and represents an important step toward addressing this critical gap in cancer disparity research.
We agree that imbalanced group sizes can limit statistical power, particularly when evaluating smaller effect sizes. To address this, we applied standard statistical tests such as chi-squared analysis and clearly reported all p-values, including those near significance thresholds, to maintain transparency. While we are cautious in our interpretation of results with marginal significance, the consistency of certain findings across multiple pathways (e.g., FGFR4, IGF1R, INPP4B, TGFBR2) highlights potentially meaningful patterns that merit further investigation.
In response to the reviewer’s suggestion, we have included a paragraph in the Methods and Discussion sections of the revised manuscript explicitly acknowledging this limitation and reinforcing the need for more comprehensive genomic datasets that better represent underrepresented populations. We are committed to advancing this work through future studies with larger, more balanced cohorts as data becomes increasingly available.
The revised paragraph added to the Methods section, lines 163-173, now reads as follows:
“Given the imbalance in sample sizes between H/L and NHW patients, we employed statistical approaches appropriate for unequal group sizes. Mutation frequency compar-isons between the two groups were conducted using chi-squared tests, which are robust for categorical data analysis even in the presence of unequal group sizes, provided cell counts are adequate. All p-values are reported to ensure transparency, and findings with marginal significance were interpreted cautiously. While the current dataset represents one of the few publicly available resources with clinical and genomic data on H/L patients with HCC, we recognize the potential limitations in statistical power, particularly for detecting smaller effect sizes. As such, our analyses emphasize the identification of no-table trends and candidate gene alterations that may warrant further validation in larger, more balanced cohorts in future studies.”
The revised paragraph added to the Discussion section, lines 655-668, now reads as follows:
“An acknowledged limitation of this study is the imbalance in sample sizes between the H/L and NHW patient groups, with 69 and 478 individuals respectively. This dis-crepancy reflects the broader underrepresentation of H/L populations in publicly available genomic datasets for HCC, which poses challenges for conducting statistically robust disparity-focused research. Despite the limited sample size, we identified several potentially meaningful differences in pathway-specific alterations, particularly in genes such as FGFR4, IGF1R, INPP4B, and TGFBR2. While larger effect sizes can still yield statistically significant findings in smaller cohorts, we recognize that smaller effect sizes may be underpowered in this context. We have therefore interpreted borderline associations with caution and reported all p-values transparently. Future studies with larger, more balanced cohorts and expanded genomic data for underrepresented populations will be essential to validate and extend these findings. Our work highlights the importance of continued efforts to increase diversity in cancer genomics research and the urgent need for more inclusive datasets to enable equitable precision medicine.”
MINOR COMMENTS:
Comment 3-4:
“Table 1 "Non Spanish non hispanic need to be clarified. Which ethnicity they belong to ?”.
Response:
We thank the reviewer for pointing out the need for clarification regarding the classification of ethnicity in Table 1. The “Non-Spanish; Non-Hispanic” designation used for the Non-Hispanic White (NHW) cohort is based on how ethnicity is recorded in the original datasets (cBioPortal and GENIE). Specifically, all patients in the NHW cohort were recorded as “Non-Spanish; Non-Hispanic” and identified as White in the race category within the source data. Therefore, the NHW group in our study includes individuals who self-identified as White and explicitly indicated they were not of Spanish, Hispanic, or Latino ethnicity. We have updated the table 1 with a note to ensure this classification is clearly understood.
The updated note for Table 1, lines 238-241, now reads as follows:
“*Ethnicity classification was based on self-reported data available from public databases. All individuals categorized as NHW were recorded as White in the race field and "Non-Spanish; Non-Hispanic" in the ethnicity field. No individuals in the NHW group reported any Hispanic, Latino, or Spanish ethnicity.”
Comment 5:
“The overall manuscript need language corrections to make sure the follow of the content is clear.”
Response:
We thank the reviewer for their helpful observation regarding the language and clarity of the manuscript. In response, we have carefully reviewed and edited the entire text to improve grammar, sentence structure, and overall flow. These revisions were made to enhance readability and ensure that the content is communicated clearly and effectively. All language corrections and clarifications have been marked using blue text and highlighted in yellow in the revised manuscript for easy reference. We appreciate the reviewer’s suggestion, which has contributed to improving the overall quality and presentation of the manuscript.
Comment 6:
“Regarding the ithenticate report, it shows 38% similarity which is way above the threshold. Kindly revise the manuscript to remove the similarity.”
Response:
We thank the reviewer for bringing this to our attention. We acknowledge the similarity report indicating a 38% overlap with other reports from our group. This overlap primarily stems from shared methodological descriptions, dataset descriptions, and standard language used across related studies within our research program. Nonetheless, we have carefully revised the manuscript to reduce redundancy and ensure originality in phrasing, while preserving the scientific integrity and clarity of the content. The updated version reflects these changes (blue text and highlighted in yellow), and we believe it now falls within acceptable similarity thresholds. We appreciate the reviewer’s attention to this matter and the opportunity to improve the manuscript.
We sincerely appreciate your thoughtful comments once again.

Reviewer 2 Report
Comments and Suggestions for Authors
In the presented manuscript the authors aim to identify differential mutations between Hispanic/Latino and Non-Hispanic white cancer patients explaining differences in cancer progression and prognosis. To this end they rely on cBioportal and GENIE patient data, which allowed them to stratify patients based on their ethnicity and genomic profiles to, lastly, compare patient survival depending on their ethnic background and mutation status.
I think the concept behind the manuscript is quite interest and relevant for the clinic, and would represent a qualitative leap in personalized medicine, however I have several concerns about it, and I’d like to have them addressed before taking a decision.
- The number of patients in each group looks quite unbalanced which, from a statistical point of view, could represent a big problem. This, together with the low number of patients in the Hispanic/Latino cohort, raise the question of whether the observations are real or just a consequence of the probable lack of power of the study. This turns a major problem when the number of patients harbouring mutations is really low.
- Origin of datasets is not properly described. cBioportal is quite a broad concept which includes datasets from multiple sources and it's mentioned that some datasets are not analysed due to the lack of details about ethnicity.
Which ones were finally used for the study? If so, have the authors considered the potential heterogeneity in how the patient data was treated as well as the possible differences in the clinical evaluations between studies?
- What reference was used to consider that a gene belongs to the pathways studied in the manuscript?
- Some genes have dual roles in tumour progression, being able to show both tumour suppressor and oncogenic roles depending on, for instance, tumour stage. Have the authors considered such important feature when pooling genes into pathways?
Moreover, authors don’t give information about the genetic alterations used for the study. Are they considering all alterations? Are they including both gain and loss of function alterations? In the manuscript CNVs are mentioned, are these genomic alterations also included?
Simplifying tumour biology and just pool all genes from a pathway with no previous caution about their roles or nature of the genetic alteration could make very difficult any type of conclusion and might explain the lack of effect shown in all the pathways.
- Authors associate some mutations more abundant in H/L to be associated to HCC, however, I wonder whether this is due to the ethnic background and not a major driver of HCC. In such case one would expect it to be associated in some way to, for instance, tumour progression or frequency of tumour development in the population. Anyways, it would be more informative to know the frequency of the mutations in the broader H/L population to determine if these differences are exclusive for cancer patients or just reflects the genetic background of the population.
- Related to the last point, have the authors considered other confounders in the cohorts which could be very relevant for the study as, for instance, alcohol consumption, healthcare access, or other lifestyle and socioeconomic factors?
- Which software was used to perform the Kaplan-Meyer survival curves and associated statistical analysis?
Author Response
Reviewer 2 Comments - Attached Word file: Response_Reviewer_2_Comments_040425.dovx
We are pleased to resubmit our manuscript, titled “Comparative Genomic Analysis of Key Oncogenic Pathways in Hepatocellular Carcinoma Among Diverse Populations,” and are confident it will be of considerable interest to the scientific community. We have thoroughly addressed all reviewer comments and have further underscored the relevance of this important cancer disparity. This work represents one of the first genomic studies of hepatocellular carcinoma (HCC) to focus on ethnic differences, utilizing a publicly available dataset with meaningful representation of both Hispanic/Latino (H/L) and Non-Hispanic White (NHW) patients. Through analysis of mutation frequencies in key oncogenic pathways—RTK/RAS, TGF-Beta, WNT, PI3K, and TP53—we identified important molecular distinctions that may underlie disparities in disease incidence and progression. In particular, we report a higher prevalence of FGFR4, IGF1R, INPP4B, and TGFBR2 mutations in H/L patients, findings that may inform precision medicine approaches and improve clinical outcomes in underrepresented populations.
We thank you for your time and thoughtful review. Please find our detailed responses to each comment below, with corresponding revisions clearly marked in blue font and highlighted in yellow in the revised Word document.
Reviewer 2’s feedback was generally positive. The reviewer acknowledged the manuscript’s focus on identifying differential mutations between Hispanic/Latino and Non-Hispanic White patients as highly relevant to understanding disparities in cancer progression and prognosis. They noted the effective use of publicly available datasets, including cBioPortal and GENIE, which enabled the authors to stratify patients by ethnicity and genomic alterations. This approach was recognized as a valuable means to explore potential molecular drivers of observed clinical differences. Additionally, the reviewer appreciated the integration of survival analysis to examine the relationship between mutation status and patient outcomes across ethnic groups. While supportive of the study's aims and methodology, the reviewer implied that further clarity and refinement in data interpretation and presentation may be needed to fully realize the manuscript’s potential impact.
Reviewer 2 writes:
“In the presented manuscript the authors aim to identify differential mutations between Hispanic/Latino and Non-Hispanic white cancer patients explaining differences in cancer progression and prognosis. To this end they rely on cBioportal and GENIE patient data, which allowed them to stratify patients based on their ethnicity and genomic profiles to, lastly, compare patient survival depending on their ethnic background and mutation status.
I think the concept behind the manuscript is quite interest and relevant for the clinic, and would represent a qualitative leap in personalized medicine, however I have several concerns about it, and I’d like to have them addressed before taking a decision.”
We appreciate the reviewer’s thoughtful and constructive feedback, and we are pleased that the concept and clinical relevance of our study were acknowledged. We are grateful for the recognition of our efforts to explore differential mutations between Hispanic/Latino and Non-Hispanic White patients and how these genomic differences may contribute to disparities in cancer progression and prognosis. We also appreciate the reviewer’s view that this work has the potential to represent a meaningful advancement in personalized medicine by incorporating ethnicity-specific genomic insights. We have carefully considered the concerns raised and have addressed them in detail to strengthen the manuscript and ensure its clarity and scientific rigor.
Comment 1:
“The number of patients in each group looks quite unbalanced which, from a statistical point of view, could represent a big problem. This, together with the low number of patients in the Hispanic/Latino cohort, raise the question of whether the observations are real or just a consequence of the probable lack of power of the study. This turns a major problem when the number of patients harbouring mutations is really low.”
Response:
We thank the reviewer for this important comment. We fully acknowledge the imbalance in cohort sizes between the H/L group (n = 69) and the NHW group (n = 478), as well as the relatively small number of patients harboring specific mutations in the H/L group. This limitation reflects a broader challenge in cancer genomics research: the historical underrepresentation of Hispanic/Latino patients in publicly available datasets, particularly for HCC. Our study leverages one of the few existing resources that include both clinical and genomic data for H/L patients with HCC, making it a valuable—albeit imperfect—opportunity to begin addressing this disparity.
We recognize that the reduced sample size in the H/L cohort may impact statistical power, particularly for detecting small effect sizes. To mitigate this, we applied rigorous statistical testing and transparently reported p-values, including those that were marginally significant. While we interpret these findings with caution, the recurrent observation of certain pathway-specific alterations (e.g., FGFR4, IGF1R, INPP4B, and TGFBR2) in the H/L group suggests potential biological relevance that merits further investigation.
In response to the reviewer’s concern, we have included detailed paragraphs in the methods and discussion section of this limitation in the revised manuscript and emphasized the need for future studies using larger, more ethnically diverse datasets. We believe that this work represents an important first step toward identifying ethnicity-specific molecular alterations that could inform precision medicine approaches for underrepresented populations.
The revised paragraph added to the Methods section, lines 163-173, now reads as follows:
“Given the imbalance in sample sizes between H/L and NHW patients, we employed statistical approaches appropriate for unequal group sizes. Mutation frequency compar-isons between the two groups were conducted using chi-squared tests, which are robust for categorical data analysis even in the presence of unequal group sizes, provided cell counts are adequate. All p-values are reported to ensure transparency, and findings with marginal significance were interpreted cautiously. While the current dataset represents one of the few publicly available resources with clinical and genomic data on H/L patients with HCC, we recognize the potential limitations in statistical power, particularly for detecting smaller effect sizes. As such, our analyses emphasize the identification of no-table trends and candidate gene alterations that may warrant further validation in larger, more balanced cohorts in future studies.”
The revised paragraph added to the Discussion section, lines 655-668 , now reads as follows:
“An acknowledged limitation of this study is the imbalance in sample sizes between the H/L and NHW patient groups, with 69 and 478 individuals respectively. This dis-crepancy reflects the broader underrepresentation of H/L populations in publicly available genomic datasets for HCC, which poses challenges for conducting statistically robust disparity-focused research. Despite the limited sample size, we identified several potentially meaningful differences in pathway-specific alterations, particularly in genes such as FGFR4, IGF1R, INPP4B, and TGFBR2. While larger effect sizes can still yield statistically significant findings in smaller cohorts, we recognize that smaller effect sizes may be underpowered in this context. We have therefore interpreted borderline associations with caution and reported all p-values transparently. Future studies with larger, more balanced cohorts and expanded genomic data for underrepresented populations will be essential to validate and extend these findings. Our work highlights the importance of continued efforts to increase diversity in cancer genomics research and the urgent need for more inclusive datasets to enable equitable precision medicine.”
Thank you once again for your valuable comment.
Comment 2:
“Origin of datasets is not properly described. cBioportal is quite a broad concept which includes datasets from multiple sources and it's mentioned that some datasets are not analysed due to the lack of details about ethnicity.
Which ones were finally used for the study? If so, have the authors considered the potential heterogeneity in how the patient data was treated as well as the possible differences in the clinical evaluations between studies?”
Response: We thank the reviewer for raising this important point. We agree that cBioPortal is an aggregation platform that hosts numerous datasets from diverse sources, and clarification is essential regarding which datasets were used in our analysis. For this study, we included HCC datasets available through cBioPortal and GENIE that contained both genomic mutation data and clearly annotated patient-level ethnicity information, specifically identifying individuals as H/L or NHW. Datasets that lacked ethnicity data or provided incomplete clinical annotation were excluded from our analysis to ensure the reliability of our population stratification
We acknowledge the potential heterogeneity in how clinical data were collected, annotated, and evaluated across different contributing studies. To mitigate this, we focused our analyses on variables that were consistently reported across the included datasets, and we limited our comparisons to high-level clinical features and pathway-level mutation frequencies rather than study-specific endpoints. This approach helped reduce the impact of inter-study variability while still enabling the identification of potential ethnicity-specific oncogenic alterations.
In response to the reviewer’s concern, we have added a detailed description of the datasets included and the exclusion criteria applied to the Methods section of the revised manuscript. Additionally, we have expanded the Discussion to acknowledge the potential heterogeneity across studies and the limitations this may introduce. We appreciate the reviewer’s suggestion, which has helped us strengthen the clarity and rigor of our dataset reporting.
The revised paragraph added to the Methods section, lines 174-183, now reads as follows:
“As a note, for this analysis, we used publicly available HCC datasets. Only datasets that included both genomic mutation data and clearly annotated patient-level ethnicity information—specifically identifying individuals as H/L or NHW—were included. Da-tasets that lacked ethnicity data, had ambiguous classifications, or incomplete clinical annotations were excluded from the study to ensure accuracy in population stratification. This approach resulted in a final cohort of 547 patients, comprising 69 H/L and 478 NHW individuals. While the data were aggregated from multiple contributing institutions, we focused on high-level, consistently reported variables such as mutation status in key oncogenic pathways (RTK/RAS, TGF-Beta, WNT, PI3K, TP53), gender, and overall survival.”
The revised paragraph added to the Discussion section, lines 670-681, now reads as follows:
“We acknowledge that the data used in this study were aggregated from multiple sources through cBioPortal and GENIE, which may introduce heterogeneity in clinical evaluations, data annotation, and treatment protocols across contributing institutions. While this reflects a limitation inherent in working with large, multi-institutional datasets, we addressed it by including only those studies that provided both genomic and clearly defined ethnicity data. Additionally, we limited our analysis to consistently reported clinical and molecular features to reduce variability across datasets. We have also clearly outlined our inclusion criteria and data sources in the Methods section. Despite these limitations, our study represents one of the first attempts to explore ethnicity-specific oncogenic pathway alterations in HCC using available public data, and highlights the urgent need for more harmonized and inclusive genomic resources to support cancer disparities research.”
We appreciate your feedback and the opportunity to improve our work.
Comment 3:
“What reference was used to consider that a gene belongs to the pathways studied in the manuscript?”
Response: We thank the reviewer for this insightful question. All pathway genes analyzed in our study were sourced from the Integrated Pathway Mapper tool available in cBioPortal, which is based on curated oncogenic signaling pathway definitions developed by The Cancer Genome Atlas (TCGA) Research Network. This tool provides standardized gene sets for key pathways, including RTK/RAS, TGF-Beta, WNT, PI3K, and TP53, ensuring consistency and biological relevance across studies.
To clarify this, we have added a citation to the foundational TCGA publication titled “Oncogenic Signaling Pathways in The Cancer Genome Atlas” (Cell, 2018) in the revised manuscript. We have also included a new paragraph in the Methods section describing the use of the Integrated Pathway Mapper and the reference used for pathway definitions. We appreciate the reviewer’s comment, which helped us strengthen the transparency and reproducibility of our pathway analysis.
The revised paragraph added to the Methods section, lines 184-191, now reads as follows:
“The genes included in each oncogenic pathway analyzed in this study—RTK/RAS, TGF-Beta, WNT, PI3K, and TP53—were obtained using the Integrated Pathway Mapper tool available through cBioPortal [40]. This tool relies on curated pathway definitions established by The Cancer Genome Atlas (TCGA) Research Network to standardize gene sets across major signaling pathways relevant to cancer. Specifically, the pathway definitions are based on the comprehensive analysis presented in a comprehensive systematic publication [41]. Utilizing this resource ensured that our pathway-level comparisons were grounded in biologically validated and widely accepted frameworks.”.
Once again, we thank you for your constructive feedback, which has helped strengthen this manuscript.
Comment 4:
“Some genes have dual roles in tumour progression, being able to show both tumour suppressor and oncogenic roles depending on, for instance, tumour stage. Have the authors considered such important feature when pooling genes into pathways?
Response: We thank the reviewer for this important observation. We fully recognize that some genes within the pathways analyzed—such as those in the TGF-Beta and PI3K pathways—can exhibit context-dependent roles in tumorigenesis, functioning as either tumor suppressors or oncogenes depending on factors such as tumor stage, cellular context, or mutational background. While our analysis grouped genes by canonical oncogenic signaling pathways as defined by TCGA through the Integrated Pathway Mapper in cBioPortal, we acknowledge that this pathway-based approach does not fully capture the functional complexity or dual behavior of certain genes.
To address this, we have added a statement in the Discussion section acknowledging this limitation and noting that while pathway-level grouping provides a high-level overview of mutation patterns, the biological impact of individual gene alterations can vary significantly. We also emphasize that further functional validation and context-specific analysis will be critical to interpret the clinical significance of these mutations, particularly when aiming to inform precision medicine strategies.
We appreciate the reviewer’s thoughtful comment, which allowed us to more clearly reflect the complexity of oncogenic signaling in HCC.
The revised paragraph added to the Discussion section, lines 682-693, now reads as follows:
“It is important to note that several genes within the oncogenic pathways analyzed in this study—particularly those in the TGF-Beta and PI3K pathways—are known to exhibit context-dependent functions, acting as either tumor suppressors or oncogenes depending on factors such as tumor stage, cellular context, and the broader mutational landscape. While we used established pathway definitions from the TCGA-based Integrated Pathway Mapper in cBioPortal to group genes for analysis, this pathway-level approach does not fully capture the dynamic and sometimes dual roles of individual genes in tumor progression. As such, the biological and clinical implications of these mutations should be interpreted with caution. Future studies incorporating functional validation and more granular stratification by tumor stage and molecular context will be essential to fully understand the role of these genes in hepatocellular carcinoma, particularly in the context of ethnicity-specific differences.”
We thank the reviewer for highlighting this important consideration, which we have now addressed in the manuscript.
Comment 5:
Moreover, authors don’t give information about the genetic alterations used for the study. Are they considering all alterations? Are they including both gain and loss of function alterations? In the manuscript CNVs are mentioned, are these genomic alterations also included?
Simplifying tumour biology and just pool all genes from a pathway with no previous caution about their roles or nature of the genetic alteration could make very difficult any type of conclusion and might explain the lack of effect shown in all the pathways.”
Response: We thank the reviewer for this thoughtful and important comment. In response, we have clarified the nature of the genetic alterations included in our study. Specifically, we focused our analysis on somatic nonsynonymous mutations, which include missense, nonsense, frame shift insertions and deletions, splice site mutations, and translation start site mutations. These mutation types were extracted at the gene level from cBioPortal and are typically associated with potential functional impact—either gain or loss of function—depending on the gene and context.
To address the reviewer’s concern about the simplification of tumor biology through pathway-level pooling, we conducted an additional analysis evaluating the nature of mutations in the genes that showed significant or borderline significant differences between the Hispanic/Latino and Non-Hispanic White patient groups (specifically TGFBR2, IGF1R, FGFR4, and INPP4B). This analysis is now detailed in newly added Table S2, and we have included explanatory paragraphs in the Methods, Results, and Discussion sections to interpret the functional relevance of these mutations more cautiously.
The revised paragraph added to the Methods section, lines 206-215, now reads as follows:
“To better characterize the biological relevance of pathway-specific alterations, we examined the nature of somatic mutations in genes that showed statistically significant or borderline significant differences between H/L and NHW patients. Specifically, we evaluated mutation types in TGFBR2, IGF1R, FGFR4, and INPP4B, focusing on altera-tions with potential functional impact. Mutation data were extracted at the gene level from cBioPortal were just available from NHW populations and included nonsynonymous variants such as missense mutations, nonsense mutations, frame shift insertions and deletions, splice site mutations, and translation start site mutations. This classification was used to distinguish potentially deleterious mutations from those with uncertain or limited impact on protein function.”
The revised paragraph added to the Results section, lines 503-517, now reads as follows:
“To further explore the potential functional impact of the observed gene alterations, we analyzed the nature of mutations in genes that showed significant or borderline significant differences between Hispanic/Latino and Non-Hispanic White (NHW) patients. As shown in Table S2, mutations in TGFBR2, IGF1R, FGFR4, and INPP4B among NHW patients were primarily classified as nonsynonymous, with notable variation in mutation types across genes. TGFBR2 mutations in NHW patients were evenly distributed between missense mutations (50%) and splice region variants (50%), suggesting potential dis-ruption of gene splicing and protein function. IGF1R mutations consisted predominantly of missense mutations (80%), with smaller proportions of frame shift deletions (7%), nonsense mutations (7%), and splice site alterations (7%). Both FGFR4 and INPP4B mu-tations were exclusively missense in nature (100%), indicating amino acid substitutions that may impact protein activity without introducing premature stop codons or frame shifts. This mutation-type analysis supports the interpretation that these genes may contribute to pathway dysregulation in HCC through distinct mechanisms of functional alteration.”
The revised paragraph added to the Discussion section, lines 694-707, now reads as follows:
“The analysis of mutation types in genes with significant or borderline significance provided additional insight into their potential biological impact on HCC pathogenesis. As shown in Table S2, mutations in TGFBR2, IGF1R, FGFR4, and INPP4B among NHW patients were predominantly nonsynonymous, with a majority being missense muta-tions. The presence of splice region alterations in TGFBR2 and the mixture of mutation types in IGF1R (including missense, nonsense, and frame shift deletions) suggest func-tional disruption of key signaling components. In contrast, mutations in FGFR4 and INPP4B were exclusively missense, potentially affecting protein function through amino acid substitutions without introducing frameshifts or truncations. These patterns highlight the variability in the nature of mutations across pathway components and reinforce the importance of considering mutation type—not just frequency—when evaluating their contribution to oncogenic signaling. Future studies incorporating functional assays and in-depth structural modeling will be valuable for validating the impact of these specific alterations and understanding their roles in driving ethnic disparities in HCC.”
The new supplementary table title, lines 751-754, now reads: “Table S2. Nature of gene mutations within genes TGFBR2, IGF1R, FGFR4, INPP4B in Non-Hispanic White HCC patients. Mutation types include frame shift deletions, frame shift insertions, missense mutations, nonsense mutations, splice site mutations, and translation start site mutations.”
Regarding copy number variations (CNVs), we did not include CNV data in our primary analysis, as the focus of this study was on somatic mutation frequency within key oncogenic pathways. CNVs were mentioned in the second paragraph of the Results section in reference to the median fraction of genome altered (FGA), a commonly used metric for assessing chromosomal instability, as shown in Table 2. While FGA provides a high-level indicator of genomic alteration burden, individual CNVs were not specifically analyzed. We acknowledge that incorporating gene-level CNV data could offer additional insight into pathway dysregulation and enhance the biological interpretation of our findings. This represents an important direction for future studies.
We appreciate the reviewer’s comment, which prompted us to expand and strengthen the depth and clarity of our analysis and interpretation.
Comment 6:
“Authors associate some mutations more abundant in H/L to be associated to HCC, however, I wonder whether this is due to the ethnic background and not a major driver of HCC. In such case one would expect it to be associated in some way to, for instance, tumour progression or frequency of tumour development in the population. Anyways, it would be more informative to know the frequency of the mutations in the broader H/L population to determine if these differences are exclusive for cancer patients or just reflects the genetic background of the population.”
Response: We thank the reviewer for this insightful and important comment. We agree that the distinction between ethnicity-associated background variation and true cancer-driving mutations is critical for interpreting the biological relevance of our findings. Our analysis focused on somatic mutations identified in HCC tumors, and as such, germline variant data or mutation frequencies in the broader, cancer-free H/L population were not available through the datasets used (cBioPortal and GENIE). Therefore, we cannot exclude the possibility that some of the observed differences in mutation frequency may, in part, reflect underlying population-specific genetic backgrounds rather than tumor-specific selective pressures.
To address this limitation, we have added a paragraph to the Discussion section emphasizing the need for caution when interpreting ethnicity-associated mutation patterns without matched germline data or population allele frequencies. We also highlight the importance of future studies that integrate somatic and germline data from diverse populations to distinguish between true oncogenic drivers and background variants. We appreciate the reviewer’s thoughtful suggestion, which strengthens the interpretive framework of our study.
The revised paragraph added to the Discussion section, lines 708-718, now reads as follows:
“An important consideration in interpreting our findings is the potential influence of population-specific genetic background on the observed differences in mutation fre-quencies. While our analysis focused exclusively on somatic mutations identified in HCC tumors, we acknowledge that some of the mutations more frequently observed in H/L patients may reflect underlying germline variation or broader population-specific genetic architecture rather than being direct oncogenic drivers. Unfortunately, the publicly available datasets used in this study do not include matched germline data or allele frequency information from non-cancer H/L populations, which limits our ability to fully distinguish between cancer-specific mutations and background variants. Future studies that incorporate both somatic and germline data across diverse populations will be es-sential to validate the cancer-specific relevance of these alterations.”
We appreciate the reviewer’s suggestion, which improved the discussions of our study.
Comment 7:
“Related to the last point, have the authors considered other confounders in the cohorts which could be very relevant for the study as, for instance, alcohol consumption, healthcare access, or other lifestyle and socioeconomic factors?”
Response: We thank the reviewer for raising this important point. We fully agree that factors such as alcohol consumption, access to healthcare, lifestyle behaviors, and other socioeconomic determinants can significantly influence both HCC risk and outcomes, and may act as confounding variables in interpreting ethnicity-associated genomic differences. Unfortunately, the publicly available datasets used in our analysis do not include detailed clinical metadata on these variables, limiting our ability to account for them in this study.
We have acknowledged this limitation in the Discussion section and emphasized the need for integrative studies that combine genomic data with rich clinical, environmental, and social determinant information. Such comprehensive datasets are critical for disentangling the complex interactions between genetic alterations and contextual factors that contribute to cancer disparities. We appreciate the reviewer’s thoughtful suggestion, which underscores the multidimensional nature of cancer health equity research.
The revised paragraph added to the Discussion section, lines 719-730, now reads as follows:
“Another important consideration in this study is the absence of data on potential confounding variables such as alcohol consumption, access to healthcare, lifestyle be-haviors, and socioeconomic status. These factors are well-established contributors to HCC risk, progression, and outcomes, and may partially account for the disparities observed between ethnic groups. However, the publicly available datasets utilized in this analysis do not contain comprehensive clinical or social determinant information, which restricted our ability to control for these influences. Consequently, while our results reveal dif-ferences in somatic mutation patterns between H/L and NHW patients, they should be interpreted in the context of broader, unmeasured environmental and societal variables. Future research efforts that integrate genomic data with robust clinical and sociodemo-graphic information will be critical for disentangling the complex factors driving HCC disparities and for advancing equitable precision medicine.”
Comment 8:
“Which software was used to perform the Kaplan-Meyer survival curves and associated statistical analysis?”
Response: We thank the reviewer for this question. All analyses, including Kaplan-Meier survival curves and the associated statistical tests, were performed using R statistical software. Specifically, we used the survival and survminer packages for survival analysis and visualization. This information has been added to the Methods section of the revised manuscript for clarity.
The revised paragraph added to the Methods section, lines 199-205, now reads as follows:
“Kaplan-Meier survival analyses were conducted to evaluate overall survival differences associated with pathway-specific mutations between H/L and NHW patients. All survival analyses and associated statistical tests were performed using R statistical software (version 4.3.2). The survival package was used to fit Kaplan-Meier curves and perform log-rank tests, while the survminer package was used to generate visualizations. Patients were stratified by ethnicity and mutation status for each pathway, and survival curves were compared using the log-rank test to assess statistical significance.”
We sincerely appreciate the reviewer’s thoughtful comments and constructive feedback. The insights provided were highly valuable in strengthening the clarity, rigor, and overall quality of our manuscript. Several of the points raised prompted important clarifications and additional analyses, which we believe have significantly enhanced the scientific depth and transparency of our work. We are grateful for the time and effort the reviewer dedicated to evaluating our study and for their contributions to improving this research.

Round 2
Reviewer 2 Report
Comments and Suggestions for Authors
I appreciate the author's effort to revise the manuscript. They have adressed all the raised concerns and overall improved the paper's quality and transparency